# Importance of γ-secretase in the regulation of liver X receptor and cellular lipid metabolism

Esteban Gutierrez[1], Dieter Lütjohann[2], Anja Kerksiek[2], Marietta Fabiano[1], Naoto Oikawa[1], Lars Kuerschner[3], Christoph Thiele[3], Jochen Walter[1]

**Presenilins (PS) are the catalytic components of γ-secretase complexes that mediate intramembrane proteolysis. Mutations in the PS genes are a major cause of familial early-onset Alzheimer disease and affect the cleavage of the amyloid precursor protein, thereby altering the production of the amyloid β-peptide. However, multiple additional protein substrates have been identified, suggesting pleiotropic functions of γ-secretase. Here, we demonstrate that inhibition of γ-secretase causes dysregulation of cellular lipid homeostasis, including up-regulation of liver X receptors, and complex changes in the cellular lipid composition. Genetic and pharmacological inhibition of γsecretase leads to strong accumulation of cytoplasmic lipid droplets, associated with increased levels of acylglycerols, but lowered cholesteryl esters. Furthermore, accumulation of lipid droplets was augmented by increasing levels of amyloid precursor protein C-terminal fragments, indicating a critical involvement of this γ-secretase substrate. Together, these data provide a mechanism that functionally connects γ-secretase activity to cellular lipid metabolism. These effects were also observed in human astrocytic cells, indicating an important function of γ-secretase in cells critical for lipid homeostasis in the brain.**

## Introduction

Presenilins (PSs) are the core catalytic components of γ-secretase complexes capable of cleaving C-terminal fragments (CTFs) of type-I transmembrane proteins after precedent ectodomain shedding (Selkoe & Wolfe, 2007; Strooper & Annaert, 2010; Langosch & Steiner, 2017). Interestingly, mutations within the two presenilin homologues, PS1 and PS2, are a major cause of early-onset familial Alzheimer disease (AD) (Kennedy et al, 2003; Tanzi, 2012; Guerreiro & Hardy, 2014). Furthermore, γ-secretase directly cleaves the amyloid precursor protein (APP) leading to the generation of amyloid β--peptides (Aβ) that aggregate and deposit in AD brains. Apart from

APP, γ-secretase can cleave numerous type-I membrane proteins and could thereby exert pleiotropic functions in cellular signaling, differentiation, and survival (Haapasalo & Kovacs, 2011; Jurisch-Yaksi et al, 2013; Agrawal et al, 2016; Oikawa & Walter, 2019).

γ-Secretase has additionally been linked to cellular lipid metabolism (Grimm et al, 2005; Landman et al, 2006; Nguyen et al, 2006; Liu et al, 2007; Tamboli et al, 2008; Area-Gomez et al, 2012; Kang et al, 2013; Cho et al, 2019). We and others have shown previously that PS function is critical for the endocytosis and intracellular transport of lipoprotein particles (Tamboli et al, 2008; Woodruff et al, 2016). The deletion of PS genes and expression of familial AD-associated mutations impaired cellular uptake of lipoproteins, resulting in aberrant regulation of cellular cholesterol metabolism.

Cholesterol transport and cellular lipid metabolism are considered to contribute to the progression of AD (Grimm et al, 2007; Di Paolo & Kim, 2011; Walter & van Echten-Deckert, 2013; Martin et al, 2014; Karch & Goate, 2015). This notion is strongly supported by the finding that the apolipoprotein (APO) E4 is the major and most common risk factor for late-onset AD (Corder et al, 1993). Furthermore, the pharmacological inhibition of acyl-CoA cholesterol acyltransferase (ACAT), also known as sterol O-acyltransferase (SOAT), an intracellular cholesterol acyltransferase, strongly reduced Aβ production and extracellular plaque pathology in cellular and mouse models of AD (Puglielli et al, 2001; Hutter-Paier et al, 2004). Increased levels of cholesteryl esters (CE) have also been observed in human AD brains and APP/PS1 double transgenic mice (Chan et al, 2012), and in primary neurons upon incubation with Aβ (Cutler et al, 2004).

The deletion of PS in fibroblasts was also found to be associated with increased cholesterol esterification and lipid droplet (LD) formation (Area-Gomez et al, 2012). LDs are cellular organelles which take the form of large, complex micelles with an outer monolayer composed of phospholipids and associated proteins, and a hydrophobic core composed mainly of triacylglycerols (TAGs) and CE (Kory et al, 2016; Barbosa & Siniossoglou, 2017; Henne et al, 2018).

Although these combined data indicate an intimate connection of lipid metabolism and the pathogenesis of AD, the underlying molecular mechanisms are poorly understood. Astrocytes play an

[1]Department of Neurology, University Hospital Bonn, Bonn, Germany    [2]Institute of Clinical Chemistry and Clinical Pharmacology, University of Bonn, Bonn, Germany    [3]Life and Medical Sciences Institute, University of Bonn, Bonn, Germany

Correspondence: Jochen.Walter@ukbonn.de

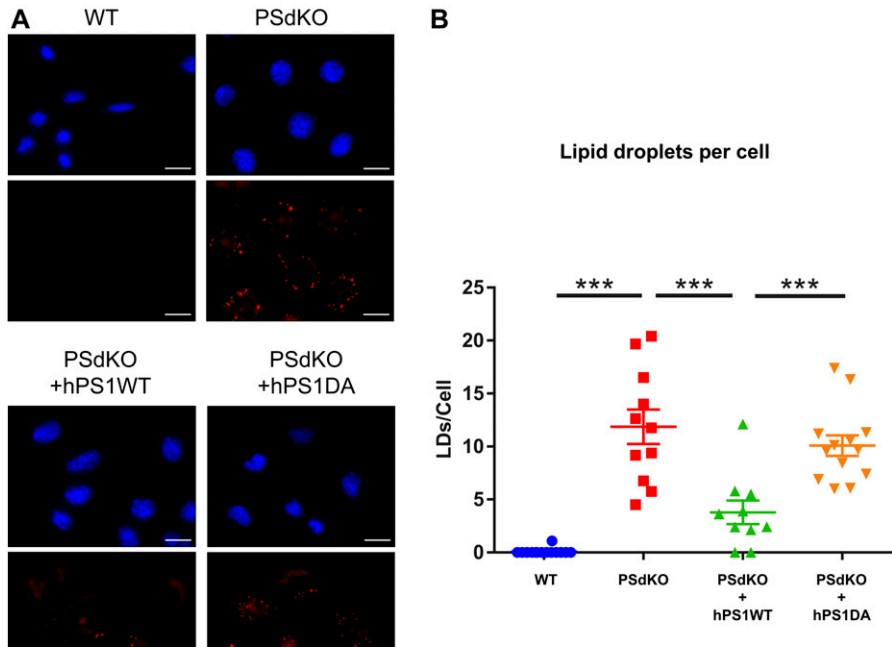

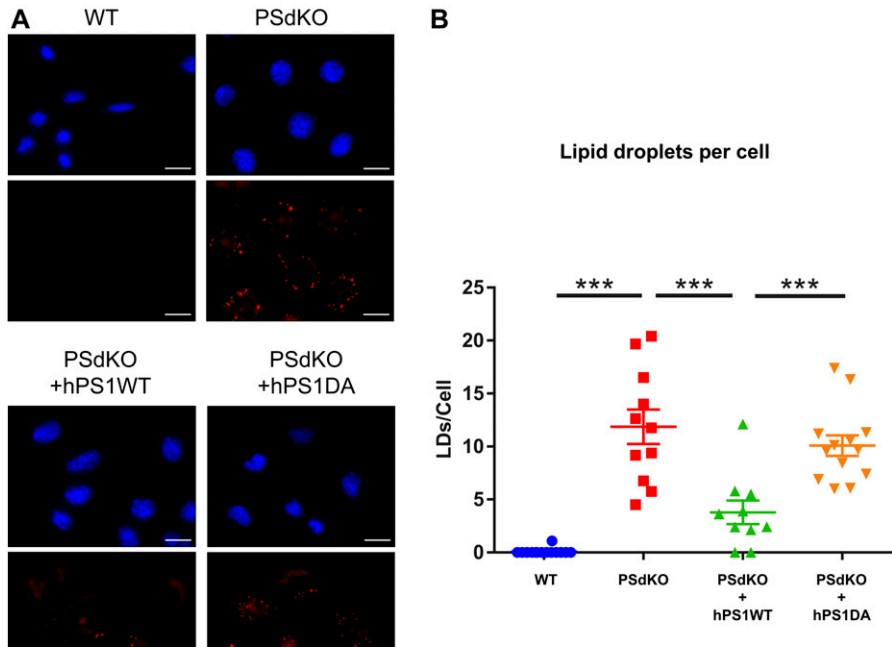

**Life Science Alliance**

**Figure 1. Presenilin-dependent accumulation of lipid droplets (LDs).**
**(A)** Representative fluorescence microscopy images of MEFs from wild-type (WT) and PS1/PS2 double knockout (PSdKO) mice. As controls, functional human PS1 WT or the catalytically inactive PS1 DA variant was stably expressed in PSdKO cells. Cells were co-stained with the LD540 dye and DAPI to visualize LDs and nuclei, respectively. An increase in the number of LDs was observed in cells lacking functional presenilin expression. Scale bar = 20 μm. **(B)** Quantification of the average amount of LDs per cell on the different genotypes. Data shown are average values ± SEM, n = 10–13. Significance was analyzed by one-way ANOVA and Holms–Sidak multiple comparison test. ***$P \leq$ 0.001.

important role in lipid metabolism of the brain and are the main cell type for synthesis and secretion of lipoproteins to deliver cholesterol and other lipids to other cells (Hayashi, 2011; Pfrieger & Ungerer, 2011; Mahley, 2016).

Here, we sought to analyze the role of γ-secretase and APP CTF in cellular lipid metabolism. Deletion of PS or pharmacological inhibition of γ-secretase led to strong accumulation of cellular LDs that was associated with decreased levels of CE but elevated levels of acylglycerols. Accumulation of the APP CTF promoted LD formation. Moreover, inhibition of γ-secretase resulted in activation of nuclear liver X receptor (LXR) and increased transcription of respective target genes, including the sterol regulatory element-binding transcription factor 1 (SREBF1), the ATP-binding cassette transporter A1 (ABCA1), and LXRβ. These data provide a functional link between γ-secretase activity and cellular lipid metabolism that could contribute lipid homeostasis in the brain and to the pathogenesis of AD.

## Results

### Accumulation of LDs upon functional impairment of γ-secretase

To assess the role of γ-secretase in cellular lipid homeostasis, we first analyzed LD levels in MEFs of wild-type (WT) and presenilin (PS) 1 and 2 double knockout (PSdKO) mice. Staining with the fluorescent lipid droplet–specific dye LD540 revealed a substantially increased number of LDs in MEFs lacking PS expression (PSdKO) as compared with WT MEFs (Fig 1A and B). As additional controls, PSdKO were stably transfected with human wild-type PS 1 (PSdKO + hPS1WT) or a catalytically inactive variant of human PS1 carrying the

D257A mutation (PSdKO + hPS1DA). Importantly, reexpression of functional PS1 (PSdKO + hPS1WT) attenuated LD accumulation. The inactive PS1 (PSdKO + hPS1DA) did not prevent the increase in LD content, suggesting that the accumulation of LDs observed in PSdKO cells is caused by loss of γ-secretase activity. To test this, WT MEFs were incubated in the presence or absence of the pharmacological γ-secretase inhibitor N-[N-(3,5-Difluorophenacetyl)-L-Alanyl]-(S)-Phenylglycin t-Butylester (DAPT) (Fig 2A and B). We also used two astrocyte cell models, given that these are the main cells involved in the production of cerebral cholesterol (Pfrieger & Ungerer, 2011). Treatment with DAPT led to a strong increase in LD content in both the H4 astroglioma cell line (Fig 2C and D) and primary human astrocytes (Fig 2E and F). These observations strongly suggest that PS-dependent γ-secretase activity plays an important role in the metabolism of LDs.

### Loss of γ-secretase activity alters the levels of CEs and acylglycerols

Because LDs mainly contain CEs and TAGs, we performed lipid analyses of cells with the different PS genotypes. Surprisingly, the level of esterified cholesterol was significantly reduced in PSdKO cells as compared with WT cells (Fig 3A). Reexpression of functional PS1 in the PSdKO cells (PSdKO + hPS1WT) normalized the levels of CEs, whereas the reexpression of the inactive PS1 mutant had no effect (Fig 3A). Pharmacological inhibition of γ-secretase with DAPT in H4 cells also decreased cellular CE levels (Fig 3B). As the fluorometric assay used for these analyses does not differentiate between cholesterol itself and other sterols, and to validate our results by an independent analytic method, we additionally performed gas chromatography-mass spectrometry

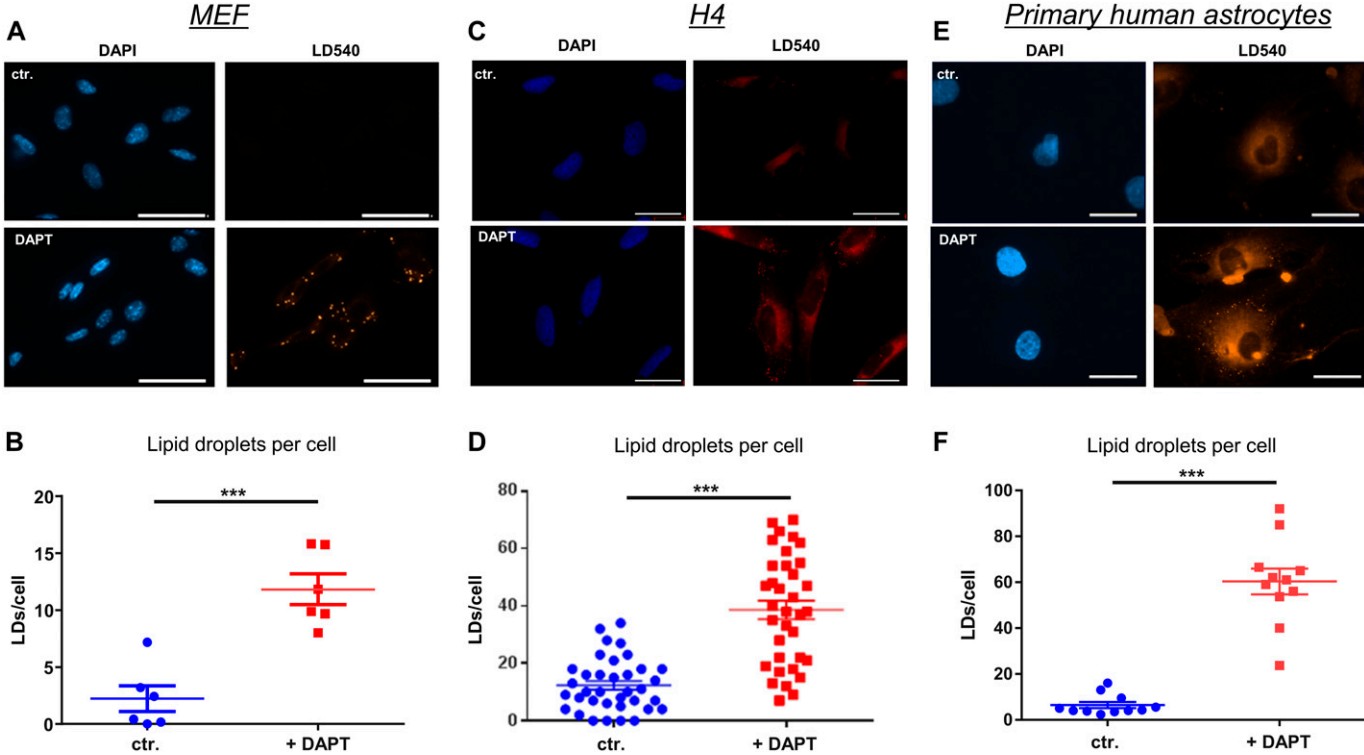

**Figure 2. Pharmacological inhibition of γ-secretase activity causes accumulation of lipid droplets (LDs).**
**(A, B, C, D, E, F)** MEF (A, B), human astrocytoma H4 cells (C, D), and primary human astrocytes (E, F) were cultured in the absence or presence of the γ-secretase inhibitor DAPT (10 $\mu$M for MEF and H4, 5 $\mu$M for primary astrocytes), and then co-stained with LD540 and DAPI. Scale bar = 20 $\mu$m. **(B, D, F)** Quantification of LDs in MEF (B; n = 6), H4 (D; n = 35), and primary human astrocytes (F, n = 11). **(B, D, F)** Data shown are average values ± SEM. Significance was analyzed by unpaired $t$ test with Welch's correction. Inhibition of γ-secretase causes a substantial increase in the number of LDs in the different cell types. ***$P \leq 0.001$.

(GC-MS) and gas chromatography-flame ionization detection (GC-FID) (Fig 3C). Analysis by GC-FID confirmed the significant decrease in CEs in PS-deficient cells. Again, reexpression of PS1 WT, but not the inactive PS1 variant, partially normalized CE levels. GC-MS analysis further showed that cells lacking expression of functional presenilin proteins (PSdKO and PSdKO + hPS1DA) contained significantly increased levels of the cholesterol precursors, lathosterol, and desmosterol, suggesting up-regulation of sterol de novo synthesis. Although total cholesterol levels were not significantly affected by the PS genotype, we observed an accumulation of filipin-positive cytoplasmic vesicles in PS-deficient cells, suggesting redistribution of free cholesterol upon deletion of PS (Fig S1). Some, but not all of the filipin-positive vesicles were also stained with antibodies against Lamp2, suggesting that cholesterol partially accumulates in lysosomes and other cytoplasmic vesicles.

As cellular levels of free cholesterol were only slightly altered, we extended our analysis of cholesterol levels and sampled the conditioned media by GC-FID for cholesterol, to evaluate cholesterol secretion from cells (Fig 3D). Compared with that of WT cells, conditioned media of PSdKO cells had significantly elevated levels of cholesterol. Whereas reexpression of inactive PS1 had no effect, expression of PS1 WT partially normalized cholesterol levels in the conditioned media. In contrast to the situation within cells, the amount of CEs in conditioned media of PSdKO cells was increased

as compared with that of WT cells. Reexpression of PS1 WT, but not of the inactive PS1 DA variant, normalized the levels of CEs in the conditioned media. Together, these data on one hand point to an increased biosynthesis and secretion of cholesterol and on the other hand to lowered cholesterol esterification in cells without functional γ-secretase activity.

To further assess cholesterol esterification in PS-deficient cells, we tested the effects of two different ACAT inhibitors, Avasimibe and K604. Whereas K604 is supposed to be highly selective for ACAT1, Avasimibe also efficiently inhibits ACAT2 (Giovannoni et al, 2003). Surprisingly, K604 increased LD content in PSdKO cells and in PSdKO cells reexpressing hPS1WT (Fig S2). The mechanisms underlying these effects remain to be elucidated but might involve changes in the expression of lipid metabolic proteins and cellular signaling pathways (Shibuya et al, 2014; Ohmoto et al, 2015). In contrast, Avasimibe which inhibits both ACAT isoforms led to a significant decrease in LDs in PSdKO cells to a level comparable with that of untreated PSdKO + hPS1WT cells (Fig S2). Treatment of PSdKO + hPS1WT cells with Avasimibe led to a slight, nonsignificant decrease in the LD content. However, as also shown in Fig 1, these cells already have a low LD content without inhibition of cholesterol esterification.

To directly test cellular cholesterol esterification, we took advantage of an alkyne cholesterol tracer and its detection by click chemistry. Cells were cultured with medium containing the

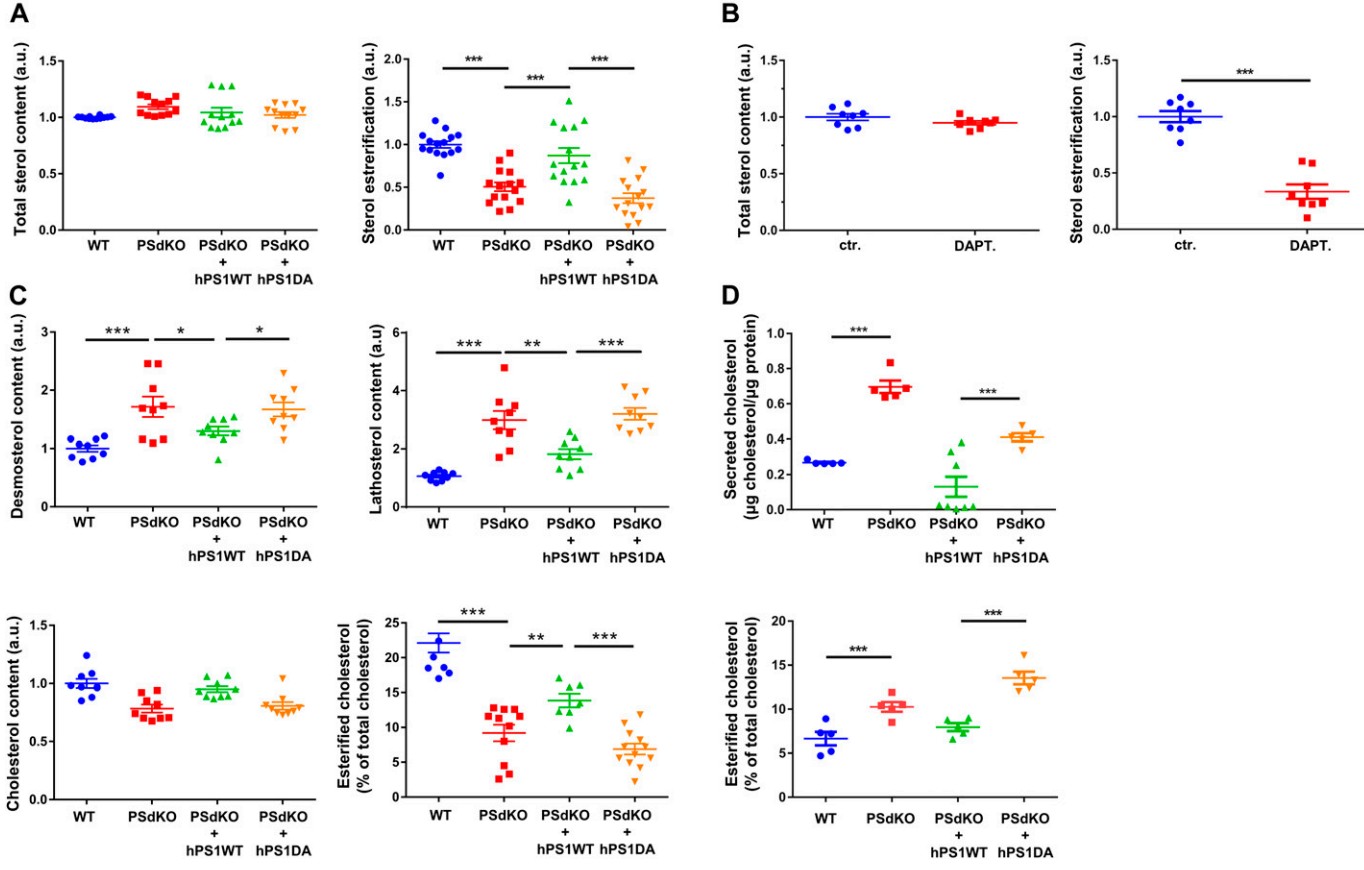

**Figure 3. Content analysis of sterol levels and sterol esterification.**
**(A, B)** Sterol and sterol esterification levels of MEFs with different PS genotypes (A), and H4 cells with and without pharmacological inhibition of γ-secretase with DAPT (B), determined by the Amplex Red Cholesterol Assay. Significantly reduced levels of cholesterol esterification were observed in cells lacking functional PS expression or upon inhibition of γ-secretase activity. Total sterol levels were not significantly. Data shown are average and individual values ± SEM, n = 12 for MEF and n = 8 for H4 cells. **(A)** In experiments with MEF cells of different PS genotypes (A), significance was analyzed by one-way ANOVA and Holms–Sidak multiple comparison test. **(B)** For comparison of cells with and without DAPT treatment (B), significance was analyzed by unpaired *t* test with Welch's correction. **(C)** Content analysis of a panel of sterols in MEFs with different PS genotypes, determined by GC-FID and GC-MS. Analyzed sterols where cholesterol precursors (lathosterol and desmosterol), as well as cholesterol and esterified cholesterol. Significantly increased levels of lathosterol and desmosterol can be observed in MEFs lacking functional PS expression, together with significantly decreased cholesterol levels. Data shown are average and individual values ± SEM, n = 9 for lathosterol, desmosterol, and cholesterol levels, and n = 7 for esterification ratio. Significance was analyzed by one-way ANOVA and Holms–Sidak multiple comparison test. **(D)** Secreted cholesterol and cholesterol esterification in culture media were analyzed by GC-FID. Significantly increased levels of secreted cholesterol can be observed in cells lacking expression of functional PS. Data shown are average and individual values ± SEM n = 4. Significance was analyzed by one-way ANOVA and Holms–Sidak multiple comparison test. *$P \leq 0.05$, **$P \leq 0.01$, ***$P \leq 0.001$.

cholesterol tracer, which carries an alkyne group at C26 (Hofmann et al, 2014). This cholesterol tracer has been shown to be uptaken, transported, and metabolized by cultured cells highly similar to cholesterol (Hofmann et al, 2014). After incubation with alkyne cholesterol, cellular lipids were isolated and probed with an azidocoumarin. After lipid separation by TLC, labeled cholesterol and cholesterol esters were detected by fluorescence imaging (Fig 4A). This experiment revealed a significantly decreased ratio of esterified cholesterol in PS-deficient cells. Reexpression of active PS1 WT partially restored cholesterol esterification, whereas inactive PS1 (DA) did not (Fig 4B).

Together, the data indicate decreased cholesterol esterification upon loss of γ-secretase activity, despite the observed increase in LD levels. Besides CEs, LDs also contain acylglycerols, in particular (TAGs) (Kory et al, 2016). The quantification of acylglycerols by an enzymatic test revealed significantly increased levels in PSdKO as compared with PS WT cells. Reexpression of human PS1 WT, but not of the PS1 DA variant, decreased acylglycerol levels significantly (Fig 4C).

To further analyze the individual species of CEs and acylglycerols, we performed tandem mass spectrometry. In line with our previous measurements (Fig 3), the level of CEs was significantly reduced in PS-deficient cells (Fig 5A). CE(18:1) represented about 50% of total CEs, and this species was significantly reduced in PSdKO cells by about 45% as compared with PS WT cells (Fig 5B). CE(18:0) and CE(20:4) that represent about 8% each of total CEs in WT cells were also reduced in PSdKO cells by about 51% and 55%, respectively (Fig 5B). Thus, a decrease in these individual CE species could contribute to the overall decreased CE content in PSdKO cells. However, CE(16:0), CE(16:1), CE(18:2), and additional minor CE species were not significantly different between WT and PSdKO cells.

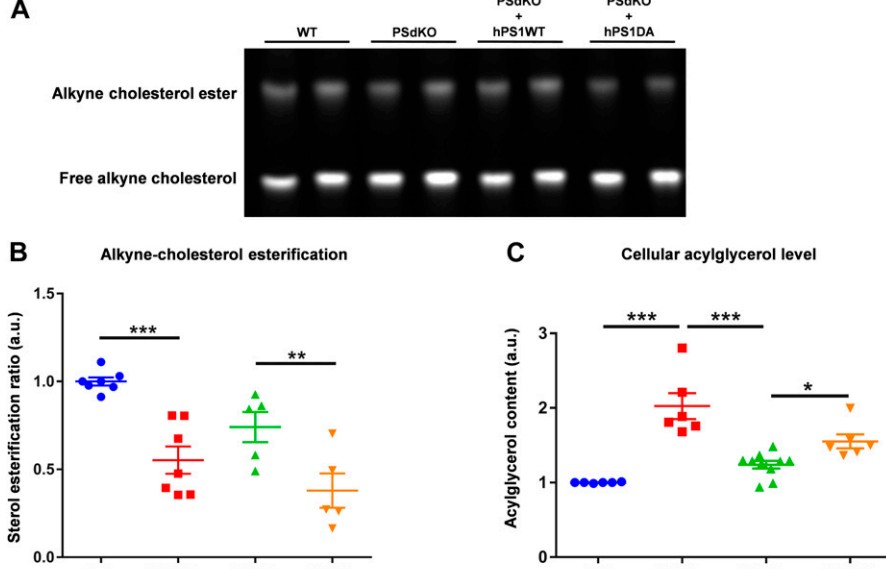

Figure 4. Analysis of sterol esterification and acylglycerol levels.
(A, B) Fluorograph of alkyne sterols click-reacted to a reporter coumarin and separated by TLC (A) and quantification of esterification (B); data shown are average and individual values ± SEM, n = 6. Significance was analyzed by one-way ANOVA and Holms–Sidak multiple comparison test. (C) Determination of cellular acylglycerol levels. Data shown are average and individual values ± SEM. n = 6. Significance was analyzed by one-way ANOVA and Holms–Sidak multiple comparison test. *$P \leq 0.05$, **$P \leq 0.01$, ***$P \leq 0.001$.

Interestingly, the total content of DAGs was significantly increased in PSdKO cells by ~65% (Fig 5C and D), whereas the content of TAGs was similar to WT cells (Fig 5E and F). The strong increase in PSdKO cells was observed for all major DAG species, including DAG(34:1), DAG(32:1) DAG(32:0), DAG(34:2), and DAG(36:2). Together, these results indicate complex changes in the cellular lipid composition in PS-deficient cells and support the contribution of increased acylglycerols to the elevated number of LDs.

### Elevated LXR activation in cells lacking functional presenilin expression

The profound changes in cholesterol and acylglycerols suggested global changes in cellular lipid metabolism upon deletion of PS or inhibition of γ-secretase. LXR is a master regulator of lipid homeostasis, and its activation could lead to an increase in cholesterol biosynthesis and efflux as well as increased synthesis of fatty acids and di- and triacylglycerols in the periphery and the brain (Oosterveer et al, 2010; Hong & Tontonoz, 2014; Courtney & Landreth, 2016). We initially analyzed LXR protein levels depending on the expression of PS (Fig 6A and B). LXR protein expression was strongly elevated in cells lacking functional PS (PSdKO and PSdKO + hPS1DA). Elevated LXR levels were observed in both nuclear and cytoplasmic fractions of PS-deficient cells. Reexpression of PS1 WT in PSdKO cells partially normalized LXR levels. These data suggest an important role of γ-secretase activity in the regulation of LXR, itself serving as a master regulator of cellular lipid metabolism.

Next, we analyzed transcription of known LXR target genes involved in cellular lipid metabolism by quantitative real-time PCR (qRT-PCR), that is, the ABCA1, known to be involved in cellular cholesterol efflux (Chawla et al, 2001), and SREBF1, a transcription factor involved in acylglycerol biosynthesis (Shao & Espenshade, 2012) (Fig 6C). mRNA expression of LXR-β itself and the two target genes was strongly elevated in PS-deficient cells. The expression of ABCA1 was, particularly, increased about 50-fold. Importantly, expression of functional PS1 WT, but not of inactive PS1 DA, strongly decreased the expression of the respective LXR target genes in PSdKO cells. The data demonstrate hyperactivation of LXR in PS-deficient cells, which could contribute to the observed effects on LD content.

### Accumulation of APP CTFs increases LD levels

γ-Secretase processes CTFs of APP and many other type-I membrane proteins. Interestingly, the β-CTF (C99) of APP has been recently described to directly interact with cholesterol (Barrett et al, 2012). Thus, we were interested in whether accumulation of APP C99 could modulate LD accumulation. To this aim, H4 cells were stably transfected with a cDNA construct encoding GFP-labeled C99 (H4 C99-GFP). C99-GFP–expressing or control H4 cells were incubated in the absence or presence of DAPT. DAPT treatment led to accumulation of C99-GFP in cytoplasmic vesicles. DAPT treatment also led to accumulation of filipin-positive cytoplasmic vesicles that mostly also contained C99-GFP, indicating co-accumulation of free cholesterol and C99-GFP in select vesicular compartments upon inhibition of γ-secretase (Fig S3). We next analyzed the association of C99-GFP and LDs (Fig 7A–D). In the absence of DAPT, LD content was low in both C99-GFP–expressing and control cells (Fig 7A and B). In C99-GFP expressing cells, the LD content significantly correlated with the expression levels of C99-GFP (Fig 7C and D). Inhibition of γ-secretase with DAPT strongly increased the fluorescence intensity of C99-GFP, demonstrating impaired processing of this γ-secretase substrate (Fig 7C). Interestingly, the number of LDs increased about threefold upon inhibition of γ-secretase (Fig 7B and C). The accumulation of LDs positively correlated with levels of C99-GFP (Fig 7D), indicating an involvement of APP C99 in the accumulation of LDs in cells with impaired activity of γ-secretase.

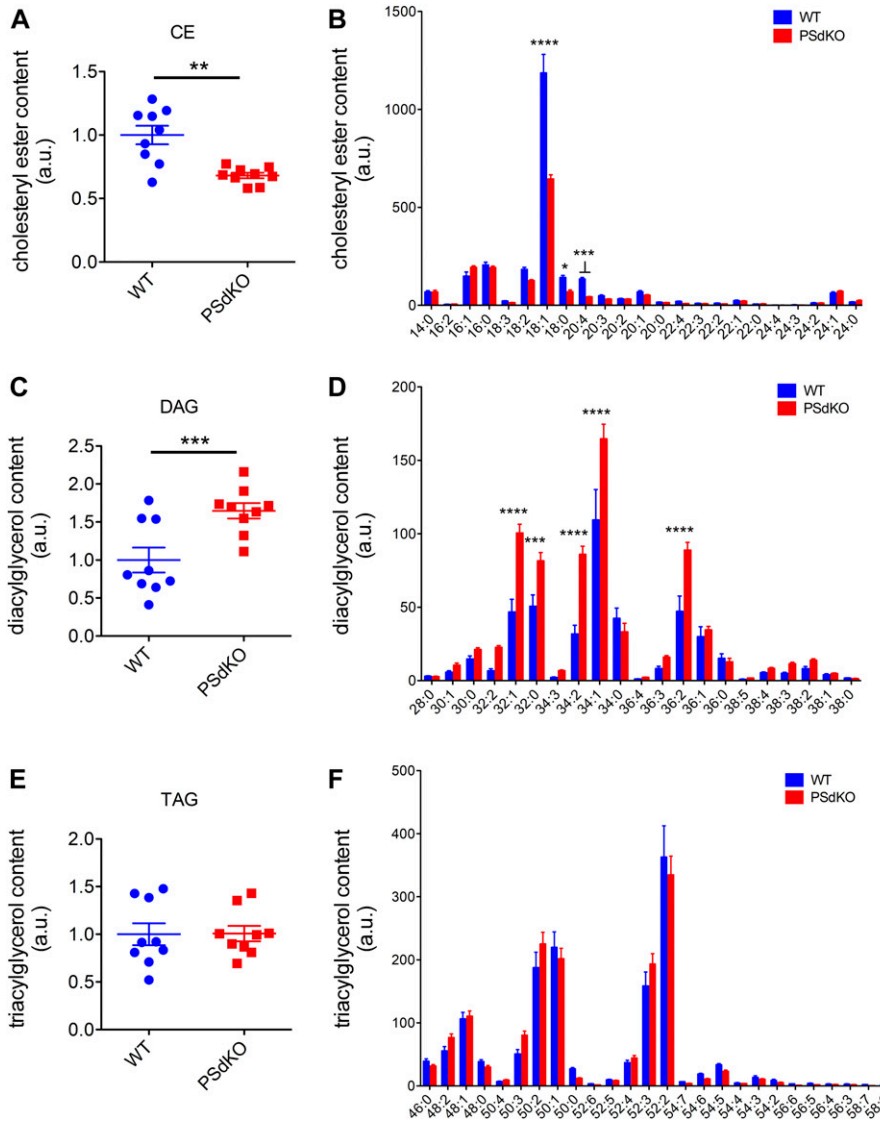

**Figure 5. Analysis of individual cholesteryl ester and acylglycerol species by tandem mass spectrometry.**
WT and PSdKO MEFs were cultured for 48 h, lysed, and the content of individual lipid species determined by tandem mass spectrometry. **(A, B, C, D)** Whereas total CE content (A) and the major individual CE species (B) were decreased, total DAG content (C) and individual DAG species (D) were significantly increased in PSdKO cells. **(E, F)** Total TAG content (E) and individual TAG species (F) were not significantly different between WT and PSdKO cells. Data shown are average and individual values ± SEM. n = 3. **(A, B, C, D, E, F)** Significance was analyzed by paired *t* test (A, C, E), and Bonferroni multiple comparison test (B, D, F). *$P \leq$ 0.05, **$P \leq$ 0.01, ***$P \leq$ 0.001, ****$P \leq$ 0.0001.

## Discussion

Present data demonstrate the importance of γ-secretase activity in cellular lipid metabolism. Inhibition of γ-secretase led to hyper-activation of LXR and impairment of cholesterol and acylglycerol homeostasis resulting in a strong accumulation of LDs. We also provide evidence for a contribution of the APP-CTF, an immediate γ-secretase substrate that accumulates in cells with decreased γ-secretase activity, in the observed lipid dysregulation.

More than 100 protein substrates of γ-secretase have been described. However, a biological role of the proteolytic processing has only been identified for very few proteins (Jurisch-Yaksi et al, 2013; Duggan & McCarthy, 2016; Walter et al, 2017), for example, for notch, Erb B4, CD44, and Eph receptors (Lammich et al, 2002; Georgakopoulos et al, 2006; Cheng et al, 2015; Kemmerling et al, 2017). Because for most of the cleavage products from other protein substrates of γ-secretase no biological function has been identi-fied, it has been proposed that γ-secretase function is also

important for the degradation of transmembrane CTFs that remain after ectodomain shedding of type-I membrane proteins (Kopan & Ilagan, 2004; Selkoe & Wolfe, 2007; Glebov et al, 2016). These CTFs might otherwise accumulate in cellular membranes and impair membrane dynamics and subcellular trafficking (Zhang et al, 2006; Kim et al, 2016; Lauritzen et al, 2019; Oikawa & Walter, 2019).

We previously showed that the accumulation of APP-CTFs upon inhibition of γ-secretase impaired the endocytosis of the low density lipoprotein (LDL) receptor, resulting in decreased uptake of lipoprotein particles and up-regulation of endogenous de novo biosynthesis of cholesterol (Tamboli et al, 2008). The impairment of lipoprotein endocytosis was attributed to the competitive binding of endocytosis adaptor proteins, including autosomal recessive hypercholesterolemia and Fe65, to accumulated APP-CTFs that also mediate the endocytosis of the LDL receptor (Tamboli et al, 2008). Reduced endocytosis of LDL has been observed in human-induced pluripotent stem cell–derived astrocytes that were deficient of APP or that express Swedish mutant APP, and these effects have been

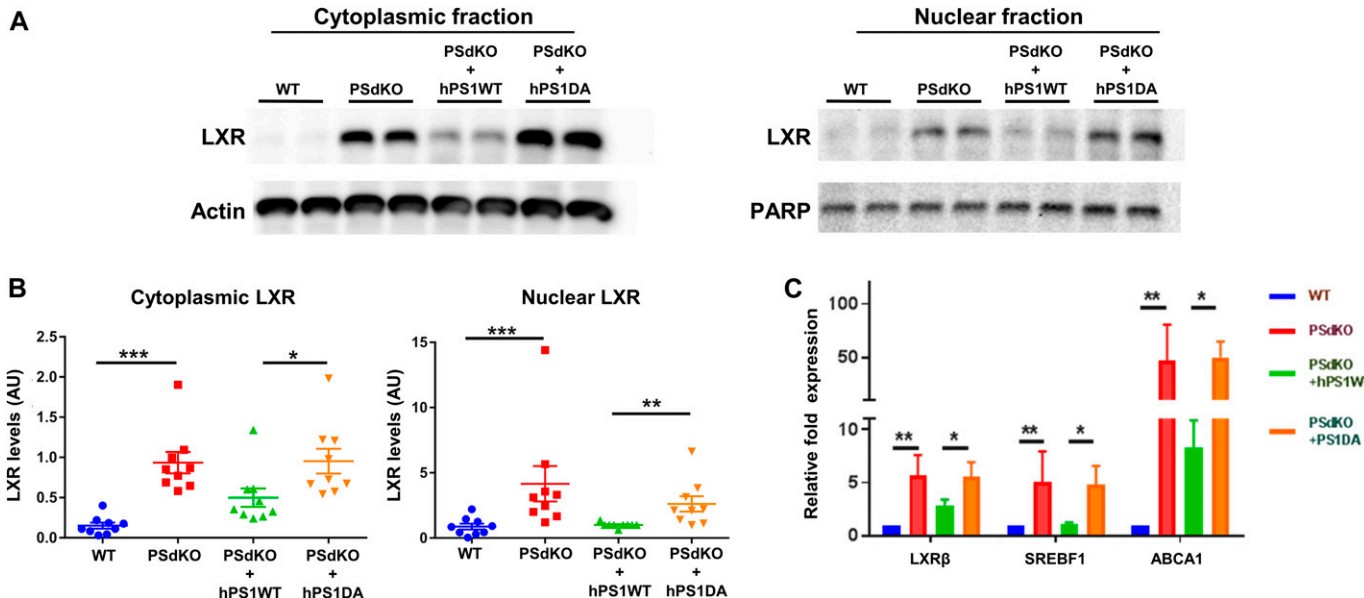

**Figure 6. PS-dependent expression of liver X receptor (LXR) and transcription of target genes.**
**(A)** LXRα/β was detected by Western immunoblotting in cytoplasmic and nuclear fractions of MEF cells with the PS genotypes indicated. **(B)** Quantification: data shown are average ± SEM (n = 9). Significance was analyzed by one-way ANOVA and Holms–Sidak multiple comparison test. LXR levels were increased in both, the nuclear and cytoplasmic fractions, in cells without expression of active PS protein. Actin and PARP were used as loading controls for the cytoplasmic and nuclear fractions, respectively. **(C)** qRT-PCR analysis of transcript levels from LXR target genes. Higher mRNA transcript levels of AbcA1 (lipoprotein and cholesterol efflux), LXRß, and SREBF1 (acylglycerol synthesis) detected on MEFs lacking PS activity. Data represent means ± SEM, n = 5. Significance was analyzed by Kruskal–Wallis nonparametric test and Dunn's multiple comparison test. *$P \leq 0.05$, **$P \leq 0.01$, ***$P \leq 0.001$.

related to altered cellular levels of full-length APP or its CTFs (Fong et al, 2018). Importantly, impaired intracellular transport of LDL has also been observed with human-induced pluripotent stem cell–derived neurons that express mutant PS (Woodruff et al, 2016). In addition to altered levels of APP and its CTFs (Tamboli et al, 2008; Woodruff et al, 2016; Fong et al, 2018), additional mechanisms, including the regulation of HMG-Co reductase by amyloid β-peptides (Grimm et al, 2007) and of LRP1 expression by the APP intracellular domain (Liu et al, 2007), have been proposed to contribute to elevated cholesterol levels in γ-secretase–deficient cells. Notably, APP and its CTF could interact with cholesterol and SREBF1 and thereby affect cholesterol homeostasis (Barrett et al, 2012; Pierrot et al, 2013; van der Kant et al, 2019).

The partial normalization of dysregulated lipid metabolism in PSdKO cells by reexpression of functional human PS1 suggests a role of PS1 in the regulation of cellular lipid metabolism. However, PS2 has also been shown to be involved in the regulation of lipid metabolism. For example, mutations associated with familial early onset AD in both, PS1 and PS2, decreased levels of phosphatidyl inositol 4,5-bisphosphate (Landman et al, 2006), and inhibition of PS2-dependent γ-secretase activity impaired sphingomyelinase activity (Grimm et al, 2005). PS1 and PS2 γ-secretase complexes are widely distributed in cellular membrane systems, including the ER, Golgi, plasma membrane, endosomal, and lysosomal comportments. However, it has been shown that PS1 complexes preferentially localize to the plasma membrane, whereas PS2 complexes showed predominant accumulation in endosomal and lysosomal

compartments and that the differential localization of PS1 and PS2 containing γ-secretase complexes contributes to the generation of distinct pools of Aβ peptides (Meckler & Checler, 2016; Sannerud et al, 2016). Thus, it will be interesting to further investigate the individual contribution of PS1 and PS2 complexes to the cellular lipid metabolism. Cell biological studies showed that PS deficiency impaired endocytosis of lipoprotein particles at the cell surface, which could alter the metabolism of cholesterol and other cellular lipids (Wood et al, 2005; Tamboli et al, 2008). In addition, PS-deficient cells showed impairment of lysosomal activity, and vesicular transport and fusion (Lee et al, 2010; Coen et al, 2012; Zhang et al, 2012; McBrayer & Nixon, 2013; Oikawa & Walter, 2019), which could also influence transport and metabolism of lipids. Indeed, stainings with filipin indicated the accumulation of free cholesterol in LAMP2-positive and other vesicular compartments in PS-deficient cells (Fig S1), thereby partially mimicking the cellular phenotype of Niemann–Pick type C (NPC) disease. In NPC disease, mutations in genes encoding NPC1 or NPC2 proteins that are both involved in vesicular cholesterol transport and esterification cause strong accumulation of free cholesterol in endolysosomal compartments and impair vesicular fusion and the sensing of cholesterol levels in cellular membranes (Pfeffer, 2019; Wheeler & Sillence, 2019). In turn, it has also been shown that the inhibition of cholesterol esterification by U18666A or NPC1 mutations leads to accumulation of PS1 and PS2 in cholesterol-rich late endosomal compartment (Runz et al, 2002; Burns et al, 2003). Together, these studies point to a close interaction of PS proteins and cellular lipid metabolism.

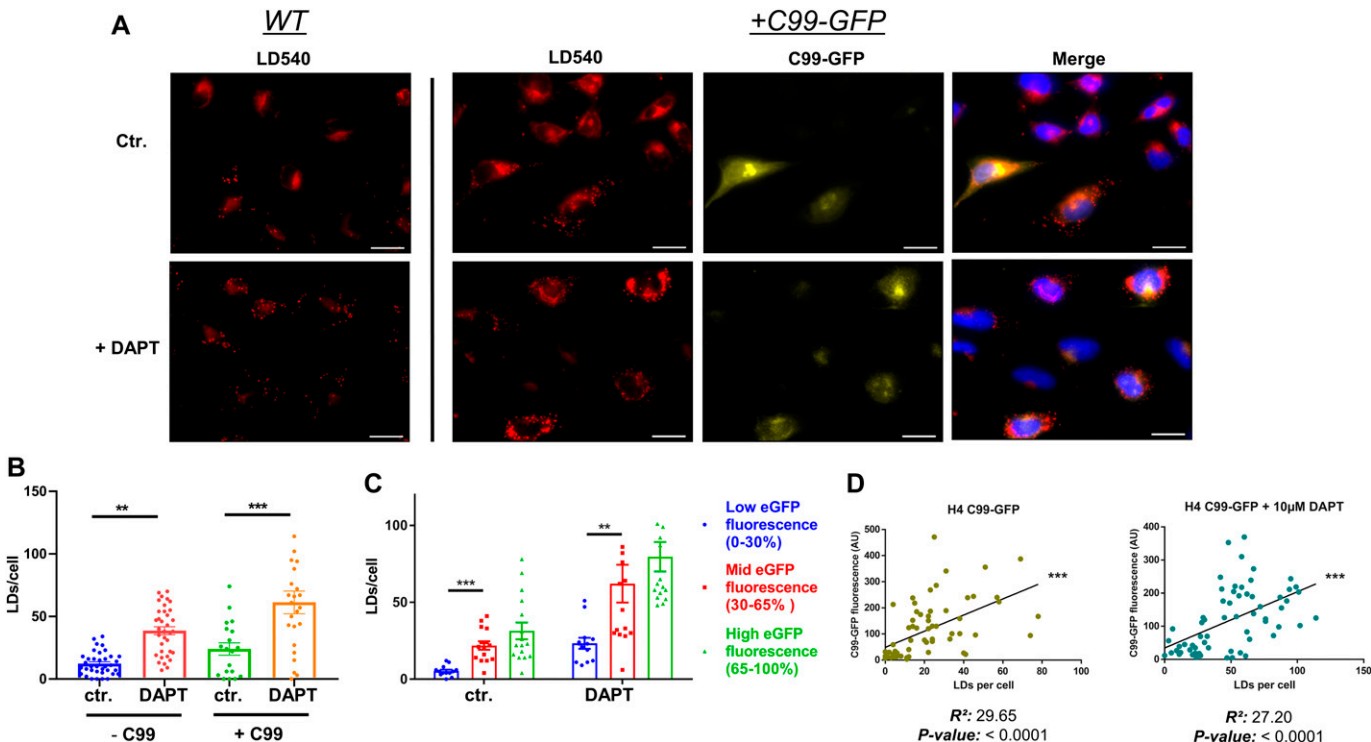

**Figure 7. Cellular levels of amyloid precursor protein–C-terminal fragments correlate with cellular lipid droplet (LD) content.**
**(A)** Representative fluorescence microscopy images of WT and C99-GFP overexpressing H4 cells with and without DAPT treatment for pharmacological inhibition of γ-secretase activity. Cells were stained for LDs with the LD540 dye. Higher levels of LDs can be observed in both WT and +C99-GFP cells after DAPT treatment, and C99-GFP overexpressing cells show higher LD levels than WT cells. In addition, cells containing elevated levels of C99-GFP display a noticeably higher number of LDs, both in the presence and absence of DAPT. Scale bar = 20 $\mu$m. **(B)** Quantification of LDs per cell (n = 16). Significance was analyzed by one-way ANOVA and Holms–Sidak multiple comparison test. **(C)** Quantification of LD levels as a function of cellular C99-GFP fluorescence. Cellular C99-GFP fluorescence levels were categorized in three different groups: Low (0–30% of max), mid (30–70% of max), and high (70–100% of max). Higher LD levels were observed in the groups with elevated C99-GFP fluorescence (n = 23). Significance was analyzed by one-way ANOVA and Holms–Sidak multiple comparison test. **(D)** Pearson linear correlation analysis of LDs and average cellular C99-GFP fluorescence. **$P ≤ 0.01$, ***$P ≤ 0.001$.

Our findings strongly support a mechanistic link between the impaired endocytosis of lipoproteins in PS-deficient cells (Tamboli et al, 2008), and the increase in LXR activation demonstrated in the present study. Taken together, these findings firmly suggest a critical involvement of LXR in the PS-dependent effect on cholesterol, LD, and acylglyceride metabolism. Cellular cholesterol can be stored in esterified form in LDs (Walther et al, 2017). Although LDs strongly accumulated in PSdKO cells or upon pharmacological inhibition of γ-secretase, detailed analysis revealed rather decreased levels of CEs. Area-Gomez et al (2012) described increased cholesteryl esterification in PS KO cells and attributed this to increased formation of mitochondria-associated ER membranes (MAMs). This discrepancy to our observations may result from different methods used to analyze cholesterol and CEs. Detailed analysis of cholesterol metabolites and esterification by mass spectrometry and alkyne cholesterol tracing strongly indicates decreased esterification of cholesterol in PS-deficient cells. Interestingly, we observed a significant increase in DAG species in PS-deficient cells, whereas TAG species were very similar between PSdKO and WT cells. It has been shown previously that DAG can promote LD formation and is found on LDs (Kuerschner et al, 2008; Cantley et al, 2013; Choudhary et al, 2018). Thus, it will be interesting to further

dissect the complex roles of γ-secretase in the regulation of lipid metabolic pathways and the dynamics of LDs.

LXR nuclear receptors are major transcriptional regulators of cellular cholesterol and TAG homeostasis (Oosterveer et al, 2010; Zhao & Dahlman-Wright, 2010; Hong & Tontonoz, 2014; Moutinho et al, 2019). Interestingly, both protein and transcript levels of LXR were significantly elevated in cells lacking γ-secretase activity. In addition, transcripts of LXR target genes, including ABCA1 and SREBF1, were also increased, thus further supporting an activation of LXR upon inhibition of γ-secretase. Activation of LXR leads to increased TAG levels, cholesterol synthesis, and efflux, as well as decreased cholesterol esterification (Zhao & Dahlman-Wright, 2010), thereby resembling lipid alterations in γ-secretase–deficient cells. Together with previous findings that the deletion of presenilins decreases the uptake of lipoprotein particles (Tamboli et al, 2008; Woodruff et al, 2016), the present data indicate that impaired endocytosis of lipoproteins from the culture media in PS-deficient cells causes hyperactivation of LXR resulting in accumulation of acylglycerols in LDs. Results were also obtained with primary human astrocytes or astrocytic cell lines, indicating a role of γ-secretase in this cell type. It will, therefore, be interesting to further investigate the specific role of γ-secretase in astrocytic lipid

metabolism in vivo. Astrocytes play a critical role in brain lipid homeostasis and represent a major source for neuronal supply with lipoproteins and cholesterol (Pfrieger & Ungerer, 2011; Courtney & Landreth, 2016). Thus, the loss of γ-secretase activity by pharmacologic inhibition or genetic mutation might affect neuronal function and contribute to neurodegeneration (Cutler et al, 2002; Di Paolo & Kim, 2011; Martin et al, 2014).

Previous studies have linked LXR to the pathogenesis of AD by altering Aβ secretion and clearance (Koldamova et al, 2003; Sun et al, 2003; Czech et al, 2007; Lefterov et al, 2007; Cui et al, 2011; Terwel et al, 2011), thereby suggesting that this transcriptional regulator of lipid metabolic pathways could represent a promising target for AD therapy and prevention (Lefterov et al, 2007; Kang & Rivest, 2012; Courtney & Landreth, 2016). Recently, the modulation of LXR and ACAT1 activity has also been shown to rescue accumulation of CE in microglia deficient in the triggering receptor expressed on myeloid cells 2 (TREM2) (Nugent et al, 2020). This immune receptor has recently been linked to AD (Guerreiro & Hardy, 2013; Jonsson et al, 2013; Colonna & Wang, 2016; Walter, 2016). Notably, TREM2 is also cleaved by γ-secretase (Wunderlich et al, 2013; Kleinberger et al, 2014), and the inhibition of γ-secretase impaired TREM2 signaling and microglia function (Glebov et al, 2016; Kemmerling et al, 2017). Thus, it is tempting to speculate that γ-secretase is also involved in the TREM2-dependent regulation of microglial lipid metabolism. The present data demonstrate that impairment of γ-secretase activity and APP-CTF accumulation lead to dysregulation of LXR and cellular lipid metabolism and, thus, indicate functional links between intramembrane proteolysis and cellular lipid homeostasis, two critical processes in the regulation of cellular homeostasis.

The modulation or inhibition of γ-secretase is explored for therapeutic targeting in Alzheimer's disease and certain cancers (Golde et al, 2013; Strooper & Chávez Gutiérrez, 2015; Medoro et al, 2018; Krishna et al, 2019; Merilahti & Elenius, 2019). Thus, it will be important to further understand the molecular mechanisms that contribute to PS-dependent effects on lipid metabolism in different cell types and the potential physiological and pathophysiological implications.

# Materials and Methods

### cDNA constructs, antibodies, and materials

The APP C99-GFP (Kaether et al, 2006) and human PS1 wild-type (hPS1WT) and the inactive hPS1D385A (hPS1DA) variant (Tamboli et al, 2008) constructs have been described previously. Filipin was obtained from Sigma-Aldrich (F9765). Primary antibodies against poly (ADP-ribose) polymerase (sc-1561; Santa Cruz Biotechnology), β-actin (A-5441; Sigma-Aldrich), liver X receptor α/β (sc-1000; Santa Cruz Biotechnology), LAMP-2 (ABL-93; Developmental Studies Hybridoma Bank), secondary horseradish-conjugated antimouse (A-9046; Sigma-Aldrich), antirabbit (A-9169; Sigma-Aldrich), and antigoat (A-5420; Sigma-Aldrich) antibodies, and secondary Alexa488-conjugated antirat (A-11006; Sigma-Aldrich) were used according to the instructions of the respective manufacturer. Standard reagents were obtained from Sigma-Aldrich unless otherwise indicated.

### Cell culture

Wild-type (WT) and PS1/PS2 double knockout (PSdKO) MEFs have been previously described (Bentahir et al, 2006). MEFs and H4 cells were cultured in DMEM supplemented with 10% fetal calf serum (PAN) and 1% penicillin and streptomycin (vol/vol) (Life Technologies). Cells were incubated at 37°C, 95% humidity, with 5% $CO_2$ atmosphere. Primary human astrocytes were cultured under similar conditions to the MEFs and H4 cells.

γ-Secretase activity was inhibited by culturing the cells with N-[N-(3,5-Difluorophenacetyl)-L-Alanyl]-(S)-Phenylglycin t-Butylester (DAPT) (Sigma-Aldrich) at the indicated concentration for at least 5 d before analysis.

### Cell fractionation and Western immunoblotting

Cell fractionation was carried out as described previously at 4°C (Kemmerling et al, 2017). Briefly, cells were washed three times in PBS and then incubated in hypotonic buffer (10 mM Tris Cl, pH 7.5, 10 mM NaCl, 0.1 mM EGTA, 25 mM ß-glycerophosphate, 1 mM DTT, and 1× complete proteinase inhibitor [Roche]) on ice for 15 min, followed by passing the lysed cells 15 times through a 0.6-mm cannula. Homogenates were centrifuged at 300g for 5 min, and the resulting supernatant (S1) and pellet (P1) were processed separately. P1 was resuspended in 100 μl buffer (25% glycerol, 20 mM Hepes, pH 7.9, 0.4 M NaCl, 1 mM EDTA, pH 8.0, 1 mM EGTA, 25 mM ß-glycerophosphate, 1 mM DTT, and 1× complete proteinase inhibitor [Roche]) and rocked ice for 20 min. This suspension was centrifuged at 16,000g for 15 min to obtain the nuclear fraction in the supernatant. To obtain cytosolic and membrane fractions, the first supernatant (S1) was centrifuged at 16,000g for 60 min. The resultant supernatant (S2) contained the cytoplasmic fraction. The pellet (P2) was resuspended in 50 μl STEN lysis buffer (50 mM Tris HCl, pH 7.6, 150 mM NaCl, 2 mM EDTA, 1% IGEPAL, 1% Triton X-100, and 1× complete proteinase Inhibitor [Roche]), incubated on ice for 10 min, and centrifuged at 16,000g for 15 min. The recovered supernatant is the membrane fraction. After fractionation, proteins were separated by SDS–PAGE and then detected by Western immunoblotting and ECL imaging.

### GC-FID and GC-MS

Cultured cells were harvested by scraping and centrifugation, washed three times with ice-cold PBS, and shock-frozen in liquid nitrogen. Samples were stored at –80°C until analysis. Absolute cellular levels of cholesterol were measured by GC-FID using 5a-cholestane as internal standard (Mackay et al, 2014) and the cholesterol precursors lathosterol and desmosterol by GC-MS with epicoprostanol as internal standard as described previously (Šošić-Jurjević et al, 2019). The degree of esterification for cholesterol was calculated from the difference in cholesterol levels with (total cholesterol) and without alkaline hydrolysis (free cholesterol) using GC-isotope dilution mass spectrometry ($26.26.26.27.27.27-^2H_6$-cholesterol as internal standard). Similar procedures were performed for measurement of absolute sterol levels or degree of cholesterol esterification from cell supernatants after direct extraction of sterols by chloroform/methanol (2:1; vol/vol).

## Tandem mass spectrometry

Cells were cultured in six-well plates. To each multiwell, 500 $\mu$l of extraction mix (CHCl$_3$/MeOH 1/5 containing internal standards: 210 pmol PE(31:1), 396 pmol PC(31:1), 98 pmol PS(31:1), 84 pmol PI(34:0), 56 pmol PA(31:1), 51 pmol PG (28:0), 28 pmol CL(56:0), 39 pmol LPA (17:0), 35 pmol LPC(17:1), 38 pmol LPE (17:0), 32 pmol Cer(17:0), 99 pmol SM(17:0), 55 pmol GlcCer(12:0), 14 pmol GM3 (18:0-D3), 359 pmol TG(47:1), 111 pmol CE(17:1), 64 pmol DG(31:1), 103 pmol MG(17:1), 724 pmol Chol(d6), and 45 pmol Acyl-Car(15:0)) were added, and the entire plate was sonicated for 2 min. The cell suspension was collected into a tube followed by centrifugation at 20,000$g$ for 2 min. The supernatant was transferred into a fresh tube and 200 $\mu$l chloroform and 800 $\mu$l 1% AcOH in water were added, and the sample was briefly shaken and spun for 2 min at 20,000$g$. The upper aqueous phase was removed and the entire lower phase transferred into a new tube and evaporated in the speed vac (45°C, 10 min). Spray buffer (500 $\mu$l of 8/5/1 2-propanol/MeOH/water and 10 mM ammonium acetate) was added, the sample sonicated for 5 min, and infused at 10 $\mu$l/min into a Thermo Q Exactive Plus spectrometer equipped with the HESI II ion source for shotgun lipidomics. MS1 spectra (resolution 280,000) were recorded in 100 m/z windows from 250 to 1,200 m/z (pos.) and 200 to 1,700 m/z (neg.) followed by recording MS/MS spectra (res. 70,000) by data-independent acquisition in 1 m/z windows from 200 to 1,200 (pos.) and 200 to 1,700 (neg.) m/z. Raw files were converted to .mzml files and imported into and analyzed by LipidXplorer software using custom mfql files to identify sample lipids and internal standards. For further data processing, absolute amounts were calculated using the internal standard intensities followed by calculation of mol% of the identified lipids.

## Assays for quantification of total sterols, acylglycerols, and proteins

Where indicated, total cellular sterol levels were also determined by the Amplex Red Cholesterol Assay (Molecular Probes) according to the manufacturer's instructions. Here, the cells were lysed in radioimmunoprecipitation (RIPA) buffer (1% NP-40, 0.1% SDS, 50 mM Tris–HCl, pH 7.4, 150 mM NaCl, 0.5% sodium deoxycholate, and 1 mM EDTA) for 10 min on ice. An excitation wavelength of 530 ± 10 nm and emission wavelength of 590 ± 10 nm were used for detection. RIPA lysis buffer, accordingly diluted, was used as a negative and H$_2$O$_2$ (Molecular probes) as positive control. The cholesterol standard curve was determined with the cholesterol/CE mixture supplied with the assay kit.

Acylglycerol content was determined in cell lysates using an enzymatic lipase-based kit (Triglyceride Quantification Colorimetric/Fluorometric Kit; BioVision) according to the instructions of the manufacturer.

Analysis of protein concentration was conducted using the Pierce BCA Protein Assay Kit following the provided indications by the manufacturer (Thermo Fisher Scientific).

## Fluorescence microscopy

Cells were cultured on uncoated glass coverslips and fixed with 4% paraformaldehyde in PBS for 10 min. LDs were visualized by staining with LD540 as described previously (Spandl et al, 2009). Briefly, fixed cells on coverslips were incubated with LD540 at a concentration of 0.025 $\mu$g/ml in PBS for 10–15 min, followed by washing three times with PBS. Free cholesterol was detected by incubation of cells with filipin at a concentration of 55.5 $\mu$g/ml in PBS for 20 min. Nuclei were stained with DAPI in PBS (300 nM, 5 min). After washing once with water, the cells were mounted in ImmuMount (Thermo Fisher Scientific) and imaged using an AxioVert 200 fluorescence microscope.

For detection of LAMP-2, fixed cells were permeabilized in a Triton X-100 solution (0.25% [vol/vol] Triton X-100 in PBS) for 10 min and incubated in blocking solution (10% [wt/vol] BSA in PBS) to prevent nonspecific protein interaction with the respective antibodies. Coverslips were washed three times with PBS containing 0.125% (vol/vol) Triton-X 100 before incubation with the primary antibody for 1 h. The cells were washed again three times in PBS and incubated with secondary antibodies for 1 h. After washing again three times in PBS and once with water, coverslips were mounted on microscopy slides with ImmuMount (Thermo Fisher Scientific) and allowed to dry overnight at 4°C (36 h for LD540). Slides were analyzed with the aid of the AxioVert 200 fluorescence microscope (Zeiss), and APP C99-GFP fluorescence was determined using the ZEN microscopy software (Zeiss).

## Alkyne cholesterol tracing assay

For the detection of free and esterified alkyne cholesterol metabolized after feeding, cells were initially fed the alkyne cholesterol for 24 h at a concentration of 10 $\mu$M in full media for 24 h. The cells were harvested and lipids isolated by a biphasic extraction. Briefly, cell suspensions (900 $\mu$l) were added to 4 ml of 2:1 methanol:chloroform and bath-sonicated for 5 min. Samples were centrifuged (5 min, 4,000$g$) and the resulting supernatants supplemented with 7 ml of 6:1 H2O:chloroform, followed by vigorous mixing and centrifugation (5 min, 4,000$g$). The chloroform fraction was recovered before solvent evaporation. The lipid pellets were redissolved in chloroform. Alkyne sterols were click-reacted to 3-azido-7-hydroxycoumarin, separated by TLC, and fluorographed as described previously (Thiele et al, 2012).

## qRT-PCR

RNA was isolated using the RNeasy Mini Kit (QIAGEN), and the cDNA of the isolated RNA was produced with the RevertAid First Strand cDNA Synthesis Kit (Thermo Fisher Scientific), as indicated by the respective manufacturer's instructions. The 7300 Real-Time PCR System (Applied Biosystems) was used to detect the relative abundance of cDNA. For this purpose, master mixes for primers corresponding to the genes of interest (primer + Sybr Green PCR Master Mix [Applied Biosystems]) and for cDNA templates (cDNA in DNase-free water) were plated in triplicates on a 96-well plate. Volume fluctuations between wells were controlled with the Rox reference dye. The levels of DNA generated through the reaction were analyzed by an automatically set cycle threshold (Ct). This Ct was then used to calculate relative gene expression (2-ΔΔCt), by normalizing to two reference housekeeping genes (lactate dehydrogenase A and actin beta). The primer pairs used were QuantiTect

primers, obtained from QIAGEN. The primer sequence is not disclosed by the manufacturer.

## Statistics

Results are shown as average ± SEM. Statistical significance is analyzed by unpaired $t$ test with Welch's correction when comparing two conditions and by one-way ANOVA and Holms–Sidak multiple comparison test when comparing more than two conditions, unless indicated otherwise.

# Supplementary Information

# Acknowledgements

We are grateful to Dr B DeStrooper for providing MEF. We also thank Drs IY Tamboli, T Nguyen, and K Glebov for generation and culturing of transgenic cells and primary astrocytes. This work was supported by grants from the Deutsche Forschungsgemeinschaft to J Walter (SFB645), N Oikawa (OI60/1-1), and C Thiele and L Kuerschner (SFB-TRR 83). E Gutierrez was supported by the international graduate school LIMES.

## Author Contributions

E Gutierrez: conceptualization, data curation, formal analysis, investigation, methodology, and writing—original draft.
D Lütjohann: data curation, formal analysis, investigation, methodology, and writing—review and editing.
A Kerksiek: investigation and methodology.
M Fabiano: data curation, formal analysis, and investigation.
N Oikawa: data curation, formal analysis, funding acquisition, and investigation.
L Kuerschner: funding acquisition, validation, methodology, and writing—review and editing.
C Thiele: data curation, formal analysis, funding acquisition, investigation, methodology, and writing—review and editing.
J Walter: conceptualization, formal analysis, funding acquisition, and writing—original draft and project administration.

## Conflict of Interest Statement

The authors declare that they have no conflict of interest.

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
