## [Reviewer comments · Life Science Alliance]

Life Science Alliance

Importance of γ -secretase in the regulation of liver X receptor and cellular lipid metabolism

Esteban Gutierrez, Dieter Lütjohann, Anja Kerksiek, Marietta Fabiano, Naoto Oikawa, Lars Kuerschner, Christoph Thiele, and Jochen Walter

DOI: <https://doi.org/10.26508/lsa.201900521>

Corresponding author(s): Jochen Walter, University Hospital Bonn

Review Timeline:

Submission Date:	2019-08-08
Editorial Decision:	2019-09-16
Revision Received:	2020-03-13
Editorial Decision:	2020-04-07
Revision Received:	2020-04-16
Accepted:	2020-04-17

Scientific Editor: Andrea Leibfried

Transaction Report:

September 16, 2019

Re: Life Science Alliance manuscript #LSA-2019-00521-T

Jochen Walter
University of Bonn
Dept. of Neurology
Molecular Cell Biology
Sigmund-Freud-Str. 25
Bonn, North Rhine Westphalia 53127
Germany

Dear Dr. Walter,

Thank you for submitting your manuscript entitled "Importance of γ -secretase in the regulation of liver X receptor and cellular lipid metabolism" to Life Science Alliance. The manuscript was assessed by expert reviewers, whose comments are appended to this letter.

As you will see, while reviewer #2 supports publication of a revised version of your work here, reviewer #1 and #3 raise overlapping concerns regarding the lipid analysis performed and note that the conclusions on 'cellular' lipid metabolism can furthermore be not made due to potential cholesterol efflux. The reviewers also note some discrepancies between the various datasets.

Given the input of the reviewers, we concluded that we can only consider a revised version of your work for publication here should you be willing to re-perform the lipid analyses as requested by reviewer #1 and #3. Addressing this issue is in our view quite demanding, so please consider your options carefully. Note that we would need strong support on a revised version by reviewer #1 and #3 in order to move forward here.

Thank you for this interesting contribution to Life Science Alliance. We are looking forward to receiving your revised manuscript.

Sincerely,

B. MANUSCRIPT ORGANIZATION AND FORMATTING:

Reviewer #1 (Comments to the Authors (Required)):

In this paper, Gutierrez et al revisit the link between presenilins (PS1) and lipid metabolism in cell culture, utilizing cells lacking PS1, rescued with wildtype or catalytically-dead PS1, as well as a gamma-secretase inhibitor. They confirm previous observations from others showing that lack of PS1 causes accumulation of lipid droplets and that the loss of catalytical activity is responsible for this phenotype. Surprisingly, while they show that lipid droplet accumulation correlates with a triglyceride increase, they find that cholesterol esters are actually downregulated, contrary to previous studies. The authors also characterize the expression of several factors involved in regulation of lipid metabolism, including LXR, and confirm upregulation of this protein and downstream targets in the PS1 KO cells.

Generally, this is an interesting paper that replicates a number of previously published observations and further establishes a link between key genes involved in Alzheimer's disease and lipid metabolism. There are however some discrepancies and additional points that need to be resolved before publication. These are as follows:

1. The reduced levels of cholesterol ester in the PS1 KO are intriguing. However, the methodologies used by the authors to measure cholesteryl ester levels are not reliable enough. The alkyne cholesterol ester assay combined with click chemistry is flawed by the fact that the sterol precursor used may not behave like endogenous cholesterol. Additionally, the GC/MS approach used does not distinguish the various fatty acyl groups present on CE. The authors should profile the different CE species with appropriate LC/MS methodologies in positive mode, as done by many labs. The discrepancy could easily be explained by the use of wrong methodologies.
2. Do ACAT1 inhibitors reduce lipid droplet formation in the PS1 KO cells? If they do, it would argue that they contain cholesterol esters. Also, do inhibitors of triglyceride synthesis reduce levels of lipid droplets? This would be a great positive control.
3. Assuming that the drop in CE is true, could it reflect lysosomal accumulation of free cholesterol and inability to exit from these organelles, as observed in Niemann-Pick disease type C cells? After all, PS1 mutant cells have been described to exhibit lysosomal defects by several groups. This could be addressed with a simple filipin stain.

Minor comments:

4. Page 3: Regarding the sentence "Increased levels of cholesterol esters have also been observed in human AD brains and APP/PS1 double transgenic mice ((Chan et al, 2012; Cutler et al, 2004; Cutler et al, 2002), I do not think that the Cutler et al studies show an increase in cholesteryl esters in AD brain.
5. Page 6: In the sentence "We recently showed that the inhibition of γ -secretase and accumulation of APP C99 impairs the internalization of LDL by inhibition of LDL receptor endocytosis ((Tamboli et al, 2008)", the word "recently" should be replaced with "previously". Eleven years is a long time.

Reviewer #2 (Comments to the Authors (Required)):

The manuscript from Walter and co-workers examines novel functional aspects of the intramembrane protease family of presenilin proteins. The study has focused on the effects of presenilin-1 (PS1) on lipid metabolism and transport. The fact that PS1-dependent changes in

these pathways within astrocytes is also of particular interest. The manuscript is well-written and the data is clearly presented and supports the conclusions of the study. However, there are a few issues that the authors should consider and address.

1. Given the number of previous reports in this area, question whether the increase in lipid droplets (Fig. 1) are confirmatory. Is this the first observation of lipid droplets in presenilin ablated cells?
2. The effects of DAPT on cultured human astrocytes is compelling but this inhibitor can also affect other proteins. Has a similar increase in lipid droplets been observed, for example, in vivo for astrocytes in PS1 conditional knockout mice?
3. Can the authors speculate on the impact of PS2 on comparable lipid pathways? PS1/2 double knockout cells showed a pronounced effect and it would be of interest to know if PS1 only knockouts or transfection of knockout cells with a functional PS2 was also able to rescue the phenotype. Some discussion on this point would be helpful.
4. PS1 inhibitors have been the focus of many drug development programs and also clinical trials but the findings of the current study suggest there are likely to be serious adverse side effects. Would the authors concur that this particular therapeutic approach may impact Abeta production but at the same time cause major problems for lipid pathways. Perhaps some discussion of this point might be warranted.

Other points:

Fig. 1B - labelling on the is PS1DN and should be corrected to PS1DA

Pg. 3 - indicates "sterol regulatory element-binding protein 1 (SREBPF1)", is this meant to be SREBP1 or is the reference to the transcription factor?

Fig. 6C - similar question on the changes in protein expression as the graph indicates "SREBF-1" and is this meant to be SREBP-1?

Reviewer #3 (Comments to the Authors (Required)):

In this study, authors showed loss/inhibition of γ -secretase led to disrupted lipid homeostasis in MEFs and astrocytic cells. The lipid profiling suggested the altered lipid metabolism is associated with the aberrant activation of LXR pathway. These observations indicate an interesting connection between lipid metabolism and γ -secretase activity. However, some conclusions are not adequately supported by the data provided, and the logic behind some experimental design is confusing and requires clarification by authors.

1. Author concluded that Loss of γ -secretase activity impairs esterification of cholesterol based on the observation that PS deficiency led to significant decrease in cholesterol ester. However, decrease in cholesterol ester may be also caused by enhanced cholesterol efflux. Especially, LXR pathway is shown to be activated in PS deficient cell, which is expected to promote cholesterol efflux and reduce cellular cholesterol ester level.

2. In figure 2, author claim γ -secretase inhibitor DAPT increased lipid accumulation in H4 cells. But the representative image (C) did not match the quantitative data (D). In fig. 2c, it seems that DAPT treatment reduced LD540 staining in H4 cells.

3. The logic behind the experimental design in fig. 6 is confusing. LXR is a sterol sensor, and uptake of LDL activates LXR. Given elevated LXR activation in PS deficient cell, why would author hypothesize that impaired LDL internalization is the underlying cause? In addition, if C99 inhibits LDLR endocytosis, then adding extra LDL is unlikely to correct the defect. The WB data in fig. 6 showed big variation within PSdKO group. I would suggest author rigorously test the reproducibility of the data, and clarify the logic behind the experimental design.

4. To confirm the results from fluorescence images in Fig. 7, I would suggest author use GC-MS to quantify cholesterol, cholesterol ester, and its precursors.

Response to reviewers:

Reviewer 1:

Generally, this is an interesting paper that replicates a number of previously published observations and further establishes a link between key genes involved in Alzheimer's disease and lipid metabolism. There are however some discrepancies and additional points that need to be resolved before publication. These are as follows:

1. The reduced levels of cholesterol ester in the PS1 KO are intriguing. However, the methodologies used by the authors to measure cholesteryl ester levels are not reliable enough. The alkyne cholesterol ester assay combined with click chemistry is flawed by the fact that the sterol precursor used may not behave like endogenous cholesterol. Additionally, the GC/MS approach used does not distinguish the various fatty acyl groups present on CE. The authors should profile the different CE species with appropriate LC/MS methodologies in positive mode, as done by many labs. The discrepancy could easily be explained by the use of wrong methodologies.

Response: We performed additional tandem mass spectrometry to characterize the lipid composition in cells with WT and PS1/2 double KO (PSdKO) genotype. Consistent with our previous results, levels of total cholesteryl esters were significantly lower in PSdKO cells as compared to WT cells. In addition, we also characterized esterified fatty acids in the cholesterol esters (CE). C18:1 was the main species in CE, and was significantly reduced in PSdKO cells. Less abundant CE species (C18:0, C20:4) were also significantly reduced. However, other less abundant species, including (C14:0, C16:0, C16:1, and C24:1) were found to be unaffected by PS deficiency. These data further support an overall decrease in CE content, and also show complex changes in the fatty acid composition of CE in PSdKO cells. We also found significant increase in total diacylglycerol (DAG) levels and individual DAG species. However, total triacylglyceride levels were similar in WT and PS deficient cells. These new data are now shown in a new main figure (new Fig. 5) and discussed in the revised manuscript

2. Do ACAT1 inhibitors reduce lipid droplet formation in the PS1 KO cells? If they do, it would argue that they contain cholesterol esters. Also, do inhibitors of triglyceride synthesis reduce levels of lipid droplets? This would be a great positive control.

Response: We tested the effect of two different ACAT inhibitors, Avasimibe and K604. While K604 is supposed to be highly selective for ACAT1, Avasimibe also inhibits ACAT2. Avasimibe reduced the lipid droplet content in PSdKO cells, while K604, paradoxically, increased lipid droplet content. Similar effects were also observed in PSWT expressing cells. The results of these experiments are provided for the information of the reviewers. However, we think these results would rather complicate the present study as additional effects of these compounds on lipid metabolism, including triglyceride metabolism, cholesterol secretion, and autophagy could potentially affect the lipid droplet content. The additional lipid analyses by mass spectrometry fully support the initial finding on decreased CE content in PSdKO cells (please see point 1, and the revised manuscript).

Effects of ACAT inhibitors on lipid droplet content in PSdKO and PSWT expressing MEFs.

A) Representative fluorescence microscopy images of MEFs co-stained with LD540, specific dye for lipid droplet, and DAPI. Before fixation, cells were treated for 5 days with Avasimibe or K405 inhibitors at 10 μ M and 15 μ M concentration, respectively. Control cells were treated with DMSO.

B) Quantification of the average amount of lipid droplets per cell. Data shown are average \pm SEM. A total number of 40-50 for control and 30-35 for treated cells from triplicate readings of 3 independent experiments (n=3) was analyzed. Statistical significance was analyzed by one-way ANOVA and Holms-Sidak multiple comparison test.

3. Assuming that the drop in CE is true, could it reflect lysosomal accumulation of free cholesterol and inability to exit from these organelles, as observed in Niemann-Pick disease type C cells? After all, PS1 mutant cells have been described to exhibit lysosomal defects by several groups. This could be addressed with a simple filipin stain.

Response: We performed filipin staining. PSdKO cells showed increased filipin fluorescence in vesicular compartments as compared to WT cells, indeed suggesting accumulation of free cholesterol in these compartments. The filipin positive vesicles in PSdKO cells showed partial co-localization with Lamp2, suggesting that cholesterol accumulates in lysosomes and additional cytoplasmic vesicles.

We provide these data in the new supplementary figure S1. Our mass spectrometry measurements, however, did not show a significant increase in cholesterol levels (Fig. 3). Together, these data suggest that increased filipin staining of vesicular compartments results from a re-distribution of

cholesterol rather than overall increased cellular cholesterol levels. So, it is possible that altered subcellular transport and/or cholesterol export from vesicular compartments contribute to the impairment of cholesterol esterification.

It has been shown previously that PS deficiency could cause alterations in endosomal and lysosomal compartments likely involving impaired vesicular trafficking and fusion, but the mechanisms are not fully understood (Esselens et al. 2004, J. Cell. Biol., 166, 1041-1054; Lee et al., 2010, Cell 141, 1146-1158; Coen et al., 2014, J. Cell. Biol., 198, 23-25; Zhang et al. 2012, J. Neurosci. 32, 8633-8648;). In addition, the inhibition of cholesterol trafficking by U18 and NPC1 deficiency lead to accumulation of PS1 in filipin positive compartments (Runz, Hartmann, 2002, J. Neurosci. 22, 1679-1689). Together, these previous studies point to an important role of PS proteins in vesicular transport, and a close relation to cholesterol metabolism. We now discuss our findings and the respective literature in more detail in the revised manuscript.

Minor comments:

4. Page 3: Regarding the sentence "Increased levels of cholesterol esters have also been observed in human AD brains and APP/PS1 double transgenic mice ((Chan et al, 2012; Cutler et al, 2004; Cutler et al, 2002), I do not think that the Cutler et al studies show an increase in cholesteryl esters in AD brain.

Response: Cutler et al. (2004) showed increased levels of cholesterol and cholesterol esters in primary neurons upon treatment with Aβeta, and increased levels of cholesterol in human AD brain and APP transgenic mice. This is now corrected in the revised manuscript. We thank this reviewer for specifying this point.

5. Page 6: In the sentence "We recently showed that the inhibition of γ-secretase and accumulation of APP C99 impairs the internalization of LDL by inhibition of LDL receptor endocytosis ((Tamboli et al, 2008)", the word "recently" should be replaced with "previously". Eleven years is a long time.

Response: We changed the sentence accordingly.

Reviewer 2:

The manuscript from Walter and co-workers examines novel functional aspects of the intramembrane protease family of presenilin proteins. The study has focused on the effects of presenilin-1 (PS1) on lipid metabolism and transport. The fact that PS1-dependent changes in these pathways within astrocytes is also of particular interest. The manuscript is well-written and the data is clearly presented and supports the conclusions of the study. However, there are a few issues that the authors should consider and address.

1. Given the number of previous reports in this area, question whether the increase in lipid droplets (Fig. 1) are confirmatory. Is this the first observation of lipid droplets in presenilin ablated cells?

Response: Accumulation of lipid droplets has been shown previously, and we had mentioned this finding in the introduction. "The deletion of PS in fibroblasts was also found be associated with increased cholesterol esterification and lipid droplet formation ((Area-Gomez et al, 2012)."

We also had discussed our data in relation to the previous study in the discussion section: "While lipid droplets strongly accumulated in PSdKO cells or upon pharmacological inhibition of γ-secretase, detailed analysis revealed rather decreased levels of cholesterol esters. Area-Gomez et al. (Area-Gomez et al, 2012) described increased cholesterol esterification in PS KO cells and attributed this

increased formation of mitochondria-associated ER membranes (MAMs). This discrepancy to our observations may result from different methods used analyze cholesterol and cholesterol esters. The detailed analysis of cholesterol metabolites and esterification by mass spectrometry and alkyne cholesterol tracing strongly indicates decreased esterification of cholesterol in PS deficient cells.”

2. The effects of DAPT on cultured human astrocytes is compelling but this inhibitor can also affect other proteins. Has a similar increase in lipid droplets been observed, for example, in vivo for astrocytes in PS1 conditional knockout mice?

Response: We fully agree with the reviewer that this is a very interesting question. However, we are not aware of studies describing accumulation of lipid droplets in astrocytes with conditional deletion of PS1. We discuss this point in the revised manuscript, and mention that it will be interesting to assess the role of γ -secretase in astrocytic lipid metabolism in future studies.

3. Can the authors speculate on the impact of PS2 on comparable lipid pathways? PS1/2 double knockout cells showed a pronounced effect and it would be of interest to know if PS1 only knockouts or transfection of knockout cells with a functional PS2 was also able to rescue the phenotype. Some discussion on this point would be helpful.

Response: Previous studies showed differential subcellular distribution of PS1 and PS2 at the plasma membrane and endocytic compartments (Sannerud et al., 2016, Cell, 166: 193–208; Meckler X, Checler F 2016. *J Biol Chem* 291: 12821–12837). Thus, PS1 and PS2 could indeed exert differential effects on vesicular transport and cellular lipid metabolism. We now discuss the potential roles of PS1 and PS2 in the regulation of cellular lipid metabolism in the revised manuscript.

4. PS1 inhibitors have been the focus of many drug development programs and also clinical trials but the findings of the current study suggest there are likely to be serious adverse side effects. Would the authors concur that this particular therapeutic approach may impact Abeta production but at the same time cause major problems for lipid pathways. Perhaps some discussion of this point might be warranted.

Response: We now discuss the potential implications of our findings on therapeutic approaches in AD and cancer. “The modulation or inhibition of γ -secretase is explored for therapeutic targeting in Alzheimer’s disease and certain cancers (Krishna et al, 2019; Merilahti & Elenius, 2019; Medoro et al, 2018; Golde et al, 2013; Strooper & Chávez Gutiérrez, 2015). Thus, it will be important to further understand the molecular mechanisms that contribute to PS dependent effects on lipid metabolism in different cell types, and the potential physiological and pathophysiological implications.”

Other points:

Fig. 1B - labelling on the is PS1DN and should be corrected to PS1DA

Response: The labeling was corrected.

Pg. 3 - indicates "sterol regulatory element-binding protein 1 (SREBPF1)", is this meant to be SREBP1 or is the reference to the transcription factor?

Response: We analyzed the mRNA expression of the Sterol Regulatory Element Binding Transcription Factor 1 gene (alias Sterol Regulatory Element-Binding Protein 1). We now corrected the abbreviation and write “...sterol regulatory element-binding transcription factor 1 (SREBF1)...”

Fig. 6C - similar question on the changes in protein expression as the graph indicates "SREBF-1" and is this meant to be SREBP-1?

Response: The abbreviation SREBF1 is now consistently used in the manuscript and the figure 6c.

Reviewer #3:

In this study, authors showed loss/inhibition of γ -secretase led to disrupted lipid homeostasis in MEFs and astrocytic cells. The lipid profiling suggested the altered lipid metabolism is associated with the aberrant activation of LXR pathway. These observations indicate an interesting connection between lipid metabolism and γ -secretase activity. However, some conclusions are not adequately supported by the data provided, and the logic behind some experimental design is confusing and requires clarification by authors.

1. Author concluded that Loss of γ -secretase activity impairs esterification of cholesterol based on the observation that PS deficiency led to significant decrease in cholesterol ester. However, decrease in cholesterol ester may be also caused by enhanced cholesterol efflux. Especially, LXR pathway is shown to be activated in PS deficient cell, which is expected to promote cholesterol efflux and reduce cellular cholesterol ester level.

Response: We fully agree with the reviewer's point. Indeed, we had analyzed the levels of cholesteryl esters in the conditioned media, and found increased esterified cholesterol in the media of PSdKO cells as compared to WT cells. These data were already included in the original figure 3 of the initial manuscript. Together with our previous finding on increased expression of ABCA1 in PSdKO cells (shown in figure 5 of the previous manuscript), it is well possible that increased cholesterol secretion contributes to the lowered levels of cholesteryl esters in PSdKO cells. We now point out these findings out more explicitly, and provide a more detailed discussion on this possibility of increased cholesterol secretion in the revised manuscript.

2. In figure 2, author claim γ -secretase inhibitor DAPT increased lipid accumulation in H4 cells. But the representative image (C) did not match the quantitative data (D). In fig. 2c, it seems that DAPT treatment reduced LD540 staining in H4 cells.

Response: We apologize for this mistake. Indeed, during assembly of this figure the respective images of control and DAPT treated cells were misplaced. We corrected this now in the new figure 2 of the revised manuscript. We thank the reviewer very much for notification, and apologize again for this mistake during assembly of the figure.

3. The logic behind the experimental design in fig. 6 is confusing. LXR is a sterol sensor, and uptake of LDL activates LXR. Given elevated LXR activation in PS deficient cell, why would author hypothesize that impaired LDL internalization is the underlying cause? In addition, if C99 inhibits LDLR endocytosis, then adding extra LDL is unlikely to correct the defect. The WB data in fig. 6 showed big variation within PSdKO group. I would suggest author rigorously test the reproducibility of the data, and clarify the logic behind the experimental design.

Response: We agree with the reviewer that the data presented in figure 6 are a bit confusing. These experiments were performed to test whether PS deficient cells could still respond to exogenous LDL, which is the case. However, since LXR protein could be hardly detected in WT cells, it is difficult to compare the relative response to exogenous LDL between WT and PS deficient cells. We repeated these experiments and obtained very similar results with very low levels of LXR in WT cells. However, since this finding does not add substantially to the major points of this manuscript, and to the data already presented in the figure 5 (now new figure 6) on strongly increased levels of LXR in PS dKO cells, we decided to remove these results from the revised manuscript.

4. To confirm the results from fluorescence images in Fig. 7, I would suggest author use GC-MS to quantify cholesterol, cholesterol ester, and its precursors.

Response: We performed additional GC-MS measurements with H4 cells expressing APP-C99. However, levels of cholesterol and cholesteryl esters were not significantly different between control and C99 expressing cells.

However, we now also stained these cells with filipin, and found marginal co-localization of C99-GFP and filipin in untreated cells. The inhibition of γ -secretase with DAPT strongly increased the filipin fluorescence in cytoplasmic vesicles and the plasma membrane, and also the co-localization with C99 in the cytoplasmic vesicles. Together these data indicate that inhibition of γ -secretase leads to co-accumulation of free cholesterol and C99 in cellular membrane compartments. However, it remains to be determined whether the accumulated C99 directly interferes with cholesterol esterification.

The new data are now presented in the new supplementary figure 2, and described and discussed in the revised manuscript.

We would like to thank all reviewers for their critical and constructive comments which we found to be very helpful to improve the manuscript.

April 7, 2020

RE: Life Science Alliance Manuscript #LSA-2019-00521-TR

Prof. Jochen Walter
University Hospital Bonn
Dept. of Neurology
Molecular Cell Biology
Sigmund-Freud-Str. 25
Bonn, North Rhine Westphalia 53127
Germany

Dear Dr. Walter,

Thank you for submitting your revised manuscript entitled "Importance of γ -secretase in the regulation of liver X receptor and cellular lipid metabolism". As you will see, the reviewers appreciate the introduced changes, and we would thus be happy to publish your paper in Life Science Alliance pending final revisions:

- Please address the remaining reviewer concerns
- Please make sure that the author order in your manuscript text and within our submission system are the same
- Please use a different color (white) for the boxes in Fig S2 to increase visibility
- Please add callouts in the manuscript text to figure 5B, 5E, 5F, 7A,B
- Please adhere to alphabetical order in the legend of Figure 2; re-structure the figure if necessary
- Please mention p-values in the figure legends next to the statistical tests mentioned

A. FINAL FILES:

-- High-resolution figure, supplementary figure and video files uploaded as individual files: See our detailed guidelines for preparing your production-ready images, <http://www.life-science->

alliance.org/authors

B. MANUSCRIPT ORGANIZATION AND FORMATTING:

Thank you for your attention to these final processing requirements.

Sincerely,

Reviewer #1 (Comments to the Authors (Required)):

The authors have addressed most of the issues this referee has raised. I think that ACAT inhibition data provided in the response letter are really intriguing and should be included into the manuscript. The authors raised with this data important issues of crosstalk between the two ACAT isoforms (ACAT1/ACAT2) and lipid droplets, and possible compensatory mechanisms leading to TG upregulation when ACAT1 is specifically blocked with K-609, but not when the dual inhibitor Avasimibe is used.

Two minor issues that need to be addressed:

- Cholesteryl ester levels are known to be decreased in cells that harbor Niemann-Pick disease type C (NPC) mutations, like NPC1 mutations. The mechanism for that is because free (unesterified) cholesterol is accumulating in lysosomes from decreased egress out of these organelles and cannot be esterified in the ER. Since the presenilin KO cells show also accumulation of free cholesterol in lysosomes based on the filipin stain, similar mechanisms may be at play. The authors should discuss this possibility and cite the appropriate literature on NPC disease models.
- During the revision period, another manuscript showed accumulation of cholesteryl ester levels in a disease model relevant for Alzheimer's and ACAT1- dependency. This should be cited as well (PMID: 31902528).

Reviewer #3 (Comments to the Authors (Required)):

Most of my concerns have been properly addressed by authors. Although the underlying mechanisms through which loss of γ -secretase activates LXR pathway remains unclear, I think it can be done in the future investigations. I recommend the manuscript to be accepted for publication.

Response to reviewers:

Reviewer 1:

1. The authors have addressed most of the issues this referee has raised. I think that ACAT inhibition data provided in the response letter are really intriguing and should be included into the manuscript. The authors raised with this data important issues of crosstalk between the two ACAT isoforms (ACAT1/ACAT2) and lipid droplets, and possible compensatory mechanisms leading to TG upregulation when ACAT1 is specifically blocked with K-609, but not when the dual inhibitor Avasimibe is used.

Response: We include the data with the different ACAT inhibitors as a new supplementary figure 2, and describe and discuss these findings in the revised manuscript.

2. Two minor issues that need to be addressed:

- Cholesteryl ester levels are known to be decreased in cells that harbor Niemann-Pick disease type C (NPC) mutations, like NPC1 mutations. The mechanism for that is because free (unesterified) cholesterol is accumulating in lysosomes from decreased egress out of these organelles and cannot be esterified in the ER. Since the presenilin KO cells show also accumulation of free cholesterol in lysosomes based on the filipin stain, similar mechanisms may be at play. The authors should discuss this possibility and cite the appropriate literature on NPC disease models.

- During the revision period, another manuscript showed accumulation of cholesteryl ester levels in a disease model relevant for Alzheimer's and ACAT1- dependency. This should be cited as well (PMID: 31902528).

Response: The relation of our findings to Niemann-Pick disease type C, and lipid related effects of TREM2 variants in microglia is now discussed in the revised manuscript, and the respective literature is cited. A copy of the revised manuscript with changes tracked is provided with the resubmission.

Again, we would like to thank all reviewers for their critical and constructive comments which we found to be very helpful to improve the manuscript.

April 17, 2020

RE: Life Science Alliance Manuscript #LSA-2019-00521-TRR

Prof. Jochen Walter
University Hospital Bonn
Dept. of Neurology
Molecular Cell Biology
Sigmund-Freud-Str. 25
Bonn, North Rhine Westphalia 53127
Germany

Dear Dr. Walter,

Thank you for submitting your Research Article entitled "Importance of γ -secretase in the regulation of liver X receptor and cellular lipid metabolism". I appreciate the introduced changes and it is a pleasure to let you know that your manuscript is now accepted for publication in Life Science Alliance. Congratulations on this interesting work.

*****IMPORTANT:** If you will be unreachable at any time, please provide us with the email address of an alternate author. Failure to respond to routine queries may lead to unavoidable delays in publication.*******

DISTRIBUTION OF MATERIALS:

Again, congratulations on a very nice paper. I hope you found the review process to be constructive and are pleased with how the manuscript was handled editorially. We look forward to future exciting

submissions from your lab.

Sincerely,
